# Combined Biologic Augmentation Strategies with Collagen Patch Graft, Microfractures, and Platelet Concentrate Injections Improve Functional and Structural Outcomes of Arthroscopic Revision Rotator Cuff Repair

**DOI:** 10.3390/jcm12175694

**Published:** 2023-09-01

**Authors:** Alessandro Colosio, Andrea Bergomi, Andrea Pratobevera, Marco Paderno, Maristella Francesca Saccomanno, Giuseppe Milano

**Affiliations:** 1Department of Medical and Surgical Specialties, Radiological Sciences, and Public Health, University of Brescia, 25121 Brescia, Italygiuseppe.milano@outlook.it (G.M.); 2Department of Bone and Joint Surgery, Spedali Civili, 25121 Brescia, Italy

**Keywords:** rotator cuff, retears, revision, augmentation, platelet concentrate, patch graft

## Abstract

Background: Arthroscopic revision rotator cuff repair (ARRCR) is challenging. Biologic strategies seem to be promising. The aim was to evaluate the effectiveness of the combination of microfractures of the greater tuberosity, augmentation with collagen patch graft, and platelet concentrate injections in ARRCR. Methods: A retrospective comparative study was conducted on patients that underwent ARRCR with a minimum follow-up of two years. Patients in the augmentation group underwent ARRCR combined with microfractures, collagen patch graft, and postoperative subacromial injections of platelet concentrate. A standard rotator cuff repair was performed in the control group. Primary outcome: Constant-Murley score (CMS). Secondary outcomes: disease-specific, health-related quality of life using the Disabilities of the Arm, Shoulder, and Hand (DASH) score; assessment of tendon integrity with magnetic resonance at least six months after surgery. Significance was set at *p* < 0.05. Results: Forty patients were included. Mean follow-up was 36.2 ± 8.7 months. The mean CMS was greater in the augmentation group (*p* = 0.022). No differences could be found for DASH score. Healing failure rate was higher in the control group (*p* = 0.002). Conclusion: Biologic augmentation of ARRCR using a combination of microfractures, collagen patch graft, and subacromial injections of platelet concentrate is an effective strategy in improving tendon healing rate. Level of evidence: retrospective cohort study, level III.

## 1. Introduction

Revision rotator cuff repair (RCR) represents a real challenge for patients and surgeons as well. A wide range of options has been described, from arthroscopic repair revision up to reverse shoulder replacement. The perfect choice could not rely only on surgeons’ skills and experience, but it must be tailored on potential predictors of outcome as well. A patient’s age, symptoms, and functional demand are greatly relevant to the treatment choice, and only symptomatic retears should be considered for revision surgery. 

Failure of RCR can be related to mechanical and biological factors. Mechanical failure is mainly related to surgical technique, while biological failure relies on poor tissue quality. Furthermore, failure can be defined as retearing when mechanical stresses exceed the structural properties of degenerated tendons or those of poorly differentiated tendon-to-bone junctions. In this case, late failure is observed, after biological fixation occurred. Alternatively, failure can be a consequence of non-healing, which is related to an insufficient primary (mechanical) or secondary (biologic) tendon-to-bone fixation. Nevertheless, assessing the cause of failure is rather difficult because, in most cases, failure is due to a combination of factors. Therefore, an effective approach in revision RCR should be focused on improving both the mechanics and biology of the repair. 

Biomechanical studies ensured the effectiveness of technological advances in the use of sutures, tapes, and different repair configurations to improve primary tendon-to-bone fixation [1] in both primary and revision RCR. Similarly, new biotechnologies offer several possible options and combinations for the mechanical and biological enhancement of RCR. For instance, patch grafts are supposed to implement mechanical features [2,3], while cell therapies and growth factors (GFs) could improve the healing potential of the repaired tendons [4]. Clinical studies on primary RCR showed that patch augmentation [5], cell therapies [6], and GFs [7,8] may improve tendon healing even when administered as separate treatments. Only a few case series without a control group recently proposed an integrated approach combining collagen patches, Platelet Rich Plasma (PRP), and bone marrow aspirate concentrate (BMAC) in revision rotator cuff repair [9,10,11]. Although they showed promising functional results, these augmentation strategies have high costs, and the lack of evidence on their effectiveness limited their common use, especially as combined treatments.

The purpose of the present paper was to evaluate the effectiveness of the combination of a collagen patch graft, microfractures of the greater tuberosity, and platelet concentrate (PC) injections in an arthroscopic revision RCR (ARRCR).

The hypothesis of the study was that the combined use of tendon augmentation techniques would improve the outcome of ARRCR. 

## 2. Materials and Methods

### 2.1. Study Design

The study was designed as a retrospective comparative study on prospectively collected data from a consecutive cohort of patients. The local IRB and Ethic Committee approved the study protocol. The study was conducted according to the principles of good clinical practice and of the Declaration of Helsinki and its updated version (Tokyo 2004). 

### 2.2. Patients

All patients who underwent ARRCR for symptomatic failure of previous posterosuperior RCR were considered eligible for the study. Patients were enrolled only after accepting the invitation to enter the study and signing a consent form. Symptomatic failure was diagnosed according to clinical examination and confirmed using magnetic resonance imaging (MRI). Structural integrity was assessed on MRI and classified according to Sugaya et al. [12]

Inclusion criteria were recurrent or persistent symptoms of pain and weakness in elevation and/or external rotation associated with major structural failure (Sugaya’s type-IV and type-V [12]); age older than 18 and a minimum 2-year follow-up. Exclusion criteria were prior surgery to the affected shoulder other than primary and revision RCR, irreparable rotator cuff tear (as diagnosed at the time of revision surgery), rotator-cuff-tear arthropathy (grade >3 according to Hamada et al. [13]), infections, rheumatic or neurologic diseases involving the shoulder girdles, and worker’s compensation.

A total of 40 patients (18 males and 22 females) were included into the study. Mean age was 63.6 ± 7.3 years. Demographic data of study population are reported in Table 1. 

### 2.3. Treatment

Two groups of patients were investigated, who differed for surgical techniques of ARRCR. Each group was composed of 20 patients.

All the surgical procedures were performed in beach-chair position under general anesthesia or interscalene block, or a combination of both. After initial diagnostic arthroscopy, release of scar adhesions, and debridement of tendon edges, the tear shape and tendon mobility were assessed. Reparability of rotator cuff was defined as the possibility to reattach the tendon to the medial side of the tendon footprint without excessive tension or extensive tendon releases, such as interval slides. Hardware from previous surgery were removed whenever possible (Figure 1). 

Cuff repair was then performed according to tear pattern. A tendon-to-bone repair was accomplished, when possible, using PEEK knotted suture anchors double-loaded with #2 high-strength sutures (5.5 FT Corkscrew; Arthrex, Naples, FL, USA). Anchors were always placed at the medial edge of the tendon footprint in a single-row configuration (Figure 2). 

In L-shaped, reverse-L-shaped, U-shaped, and V-shaped tears, tendon-to-bone repair was combined with the margin convergence technique consisting of side-to-side repair with #2 high-strength sutures (FiberWire; Arthex, Naples, FL, USA). 

In the control group, cortical abrasion of the greater tuberosity was accomplished before anchor placement. In the augmentation group, cortical bone of the greater tuberosity was not abraded; after anchor placement and suture passage and before knot tying, multiple microfractures were performed using an angled arthroscopic awl for small joints onto the footprint area, between and just lateral to the suture anchors (Figure 3). 

After knot tying, all the remaining exposed area of the greater tuberosity was vented with multiple microfractures (Figure 4). 

Repair was augmented in group 2 using an extracellular matrix (ECM) made from porcine dermis (DX Reinforcement Matrix; Arthrex). The patch was carefully sized to be placed on the bursal side of the tendon and over the greater tuberosity. Two #2 high-strength sutures (FiberWire) were used to fix the graft to the rotator cuff medially; after tying the knots for medial fixation, the same sutures were passed over the patch and fixed to the greater tuberosity with two knotless PEEK anchors (4.5 mm Pushlock; Arthrex) in a suture bridge configuration (Figure 5). 

Three weekly subacromial injections of platelet concentrate (autologous conditioned plasma, ACP^®^ Double-Syringe System, Arthrex, Naples, FL, USA) were performed in patients of group 2, starting 7–10 days after surgery [14].

All patients of both groups underwent the same postoperative treatment. The operated limb was immobilized in a shoulder abduction sling for 6 weeks. The rehabilitation program started four weeks after surgery. The first phase (4 to 8 weeks post-op) focused on recovery of range of motion (ROM); the second phase (8 to 12 weeks post-op) focused on muscle strengthening in a closed kinetic chain; the third phase (12 to 16 weeks post-op) consisted of muscle strengthening exercises in an open kinetic chain, proprioceptive exercises, and postural rehabilitation of the kinetic chain.

### 2.4. Outcome Measurements

Patients were clinically evaluated once a month until the third month, then six months after surgery. A postoperative MRI was performed at 6 months. Baseline socio-demographic information and medical history were collected from medical records at the time of enrollment.

The primary outcome of the study was the functional assessment of the shoulder using the Constant-Murley score (CMS) [15]. The CMS is based on subjective (sleep, work, and recreational activities) and objective (range of motion and strength) components, adjusted for age and gender [16]. The summary score ranges from 0 (worst result) to 100 (best result). 

The secondary clinical outcome was the assessment of disease-specific, health-related quality of life using the national validated versions of the DASH (Disability of Arm, Shoulder, and Hand) questionnaire in its short version (Quick-DASH) [17]. This is a self-administered questionnaire that measures physical ability and symptoms of the upper extremity and explores the impact of functional impairment and pain on daily living tasks, as well as social and recreational activities, work, and sleep. The score of the questionnaire is based on a metric scale, ranging from 0 points (minimum disability, best result) to 100 points (maximum disability, poorest result).

A further secondary outcome was the evaluation of the structural integrity of the repaired rotator cuff on postoperative MRI. Repair integrity was measured on coronal, sagittal, and axial scans on T2-weighted sequences and classified according to the dichotomized classification of Sugaya: the repaired cuff was considered healed in Sugaya type I -III or re-torn in case of Sugaya type IV or V [18].

Primary and secondary clinical outcome measures were collected at follow-up visits by an examiner blinded to participants’ allocation.

### 2.5. Data Analysis

Sample size calculation was performed using the power analysis software G*Power v. 3.1.9.6. The sample size was calculated according to the primary outcome (CMS) and on the mean postoperative score (60.4 ± 6.7) obtained in a previous study from a population of patients with the same clinical characteristics undergoing ARRCR [19]. Based on the literature, the minimal clinically important difference (MCID) in CMS in a population of subjects undergoing rotator cuff surgery is considered to be a change of 10.4 points from baseline [20]. Based on these data, the effect size (ES) was calculated equal to 1.55. Using an a priori power analysis model and a two-tailed alternative hypothesis, given an alpha level = 0.05 and a power (1 − beta) = 0.95, a minimum sample of 12 cases for each group was calculated.

Propensity score (PS) matching was conducted for the surgical treatment between standard and augmented ARRCR. A logistic regression model was used to obtain 2 similar groups in terms of age, sex, and follow-up. A 1:1 nearest-neighbor algorithm with a caliper of 1.1 was applied to match patients using their corresponding propensity scores. PS matching was conducted using R (Version 13.0; R Development Core Team).

Data analysis was performed using SPSS Statistics 26 software (IBM, Harmonk, NY, USA).

Descriptive statistics for discrete variables were reported as mean and standard deviation as verified for normal distribution using the Shapiro–Wilk test. Otherwise, the median and interquartile range (IQR) were considered. Categorical variables were expressed as absolute frequencies and percentages.

Comparison between groups for all continuous variables at baseline and at the follow-up was carried out with the Student’s *t*-test for normally distributed data, otherwise the Mann–Whitney U-test was used. Within-group differences (baseline vs. follow-up) for continuous variables were analyzed with a paired *t*-test or with a Wilcoxon signed-rank test for data with non-normal distribution. Differences for categorical variables were assessed with a chi-squared test. Significance was considered for *p* values < 0.05.

## 3. Results

No significant differences between groups were found for baseline characteristics (Table 2). 

The mean follow-up was 36.2 ± 8.7 months (range, 24–51 months). Significant improvement was observed at follow-up compared to baseline conditions for all the outcome measures in both groups (*p* < 0.001)**.** At comparison between groups, CMS was significantly better in the augmentation group than in the control group. DASH score was lower (better) in the augmentation group, albeit the difference was not significant. Assessment of structural integrity showed a significantly lower failure rate in the augmentation group (20%) than in the control group (70%) (Table 3).

## 4. Discussion

The main finding of this study is that the combined use of biologic augmentation techniques such as collagen patch graft, microfractures, and platelet concentrate provided better functional and structural outcomes compared to the standard ARRCR. No complications were reported with the use of porcine patch grafts.

Patches had been proposed to enhance biomechanics and biology. They are composed of an acellular extracellular matrix (ECM), which is supposed to reduce the load on the repaired tendon so it can be protected during the healing phase and to induce native tissue to integrate by favoring vascularization and local cellular growth. Biomechanical studies [2,3] showed improvement in initial strength of the construct, as well as recent studies [21,22,23] that supported the clinical benefits of patch augmentation in rotator cuff repair, explained by the reduction in retear rate and pain. 

Several types of patches are available, like xenograft, allograft, and synthetic [5]. Porcine small intestine submucosa patches were the first to be proposed, but they showed very poor results and frequent complications [24], and therefore had been largely abandoned. Conversely, dermal xenograft, like the patch used in the current study, showed encouraging functional and structural results in the primary setting [25,26]. Nevertheless, human dermal allografts and synthetic grafts revealed more promising biomechanical and early clinical results than xenografts [27]. Regardless of the type of patches, longer operative time and costs are undeniable disadvantages of this procedure. However, a recent study [28] showed that a graft augmentation may represent a cost-effective procedure even in primary rotator cuff repair.

Microfractures of the greater tuberosity were also performed in the present study to increase the healing potential [29]. Arajwat et al. [6], in their meta-analysis, confirmed that bone marrow stimulation reduces the retear rate. A recent randomized clinical study [30] conducted on 69 patients who underwent a rotator cuff repair showed that small and deep bone vents (nanofractures) of the greater tuberosity halves the retear rate at a 12-month follow-up.

Lastly, the biological strategy we used to maximize tendon healing was the use of platelet concentrate. 

ACP was injected 7–10 days after surgery for two main reasons: (1) to avoid a possible dilution or washout of its effect, which may occur with arthroscopic fluid lavage [31]; (2) to allow a potentially prolonged upregulation of growth factors involved in the tendon healing cascade. Animal studies on rats have shown that application of platelet-derived growth factors on day 7 have a more pronounced effect on tendon cellular maturation and biomechanical strength than in an earlier application [32,33]. This is probably due to the fact that the inflammatory phase (capillary proliferation and collagen protein production) reaches its peak 10 days after repair [34].

The role of platelet-derived GFs in rotator cuff surgery is still a matter of debate. However, a recent systematic review of a meta-analysis [4] showed that they are effective in reducing retears, improving functional outcome scores, and reducing short-term pain. Thirteen meta-analyses were included. The authors explained that there is a conflict of conclusions between early and later studies: the former reported poorer results than the latter. Later meta-analyses seemed to have appropriate numbers, therefore the authors concluded that platelet concentrate is recommended for augmenting rotator cuff repair, and the differences in preparation, application, and consistency do not affect the outcomes.

The added value of the present study relies in the integrated approach of three different biologic augmentation strategies for revision rotator cuff repair. The rationale of our treatment consisted of a combination of GFs that may stimulate proliferation and differentiation of multipotent bone marrow cells attracted at the tendon repair site through bone vents and a scaffold that potentially provides support for in situ seeding of bone marrow cells while protecting tendon from retearing during the early postoperative period. 

Only few case series without a control group attempted an integrated approach in revision setting through the use of a collagen patch enriched with PRP and BMAC from the proximal humerus [9,10,11]. Overall functional improvement and pain reduction was reported, although one study reported up to 40% revision rate [11]. Two other studies [9,10] reported similar results. Particularly, 45% of patients achieved the minimal clinically important difference (MCID), 41% achieved the substantial clinical benefit (SCB), and 32% reached or exceeded the patient-acceptable symptomatic state (PASS) criteria for the ASES score. Although these studies do not seem to be encouraging, it must be noticed that structural failure rate in the revision of large recurrent rotator cuff retears using standard arthroscopic repair has been reported varying from 36% to 90% [35,36,37]. Accordingly, the present study showed that patients who underwent a standard rotator cuff repair reported only a 30% healing rate, as 80% of the lesions were massive. Conversely, patients who underwent an integrated approach experienced a successful healing rate approximating 80% with significantly better functional outcomes. Better functional outcome could be explained by the fact that an intact RCR maintains at least the preoperative state of fatty infiltration and muscle atrophy [38]. Skoff et al. [37] proposed revision rotator cuff reconstruction using a bridging graft composed of autogenous long head of biceps tendon (LHBT) saturated with concentrated autologous bone marrow from the iliac crest in 25 patients. The authors showed functional improvement and a 91% structural integrity at a mean follow-up of 68 months. The idea to use an autograft, such as LHBT, is surely easy and cost-effective, and augmentation with LHBT in primary RCR has been reported with the aim to reduce retear risk [39]. However, in revision RCR, the LHBT is usually no longer available to be used as scaffold augmentation, and an alternative option must be considered.

Postoperative imaging assessment of structural integrity is still a matter of debate. A recent systematic review and meta-analysis [40] showed no significant differences between ultrasound (US) and MRI for the diagnosis of rotator cuff tendon tears after prior cuff repair. Particularly, data of non-contrast MRI and MR arthrography (MRA) were pooled together, therefore it was not possible to estimate a difference between the two modalities. Although MRA is usually thought to be more sensitive, no studies compared the accuracy of the two modalities in the diagnosis of rotator cuff retears after surgery. Moreover, most clinical studies still rely on postoperative non-contrast MRI or US examination [30,36,37]. Therefore, a non-contrast MRI was performed in the present study.

This study has some limitations. First, the retrospective design impairs its external validity; further, the sample size is inadequate for a stratified analysis according to potential predictors of outcome, such as patient’s age, tear size, and alterations of rotator cuff muscles (fatty infiltration and muscle atrophy). Finally, a single short-term follow-up at a minimum of 2 years could not provide sufficient information regarding possible deterioration of anatomic and functional outcomes over time.

## 5. Conclusions

Biologic augmentation of revision RCR using a combination of microfractures, collagen patch graft, and subacromial injections of platelet concentrate is an effective strategy in improving tendon healing rate. Therefore, when considering an ARRCR, it is advisable to perform an integrated approach to maximize the healing response.

## Figures and Tables

**Figure 1 jcm-12-05694-f001:**
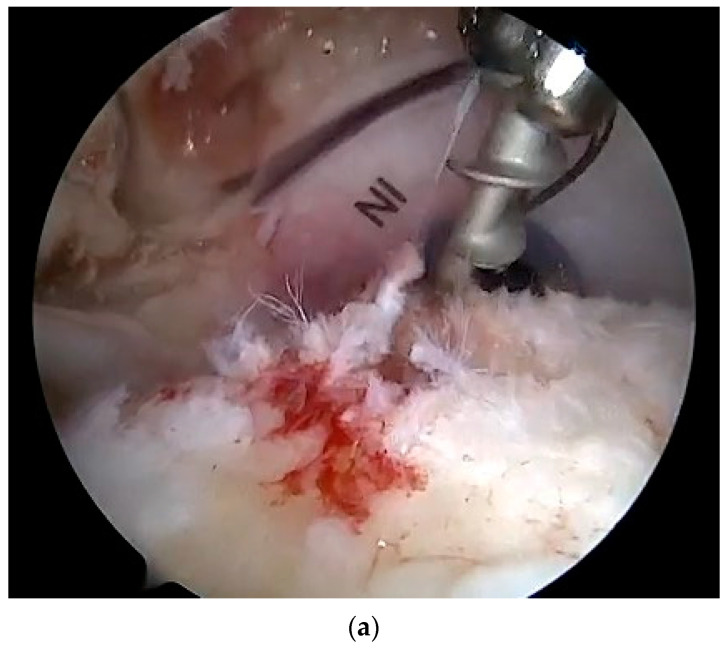
Right shoulder: scope in the posterior portal. Hardware removal attempt: a metallic anchor was used during the primary repair. A nitinol wire is used to secure the anchor (**a**) before removing it (**b**).

**Figure 2 jcm-12-05694-f002:**
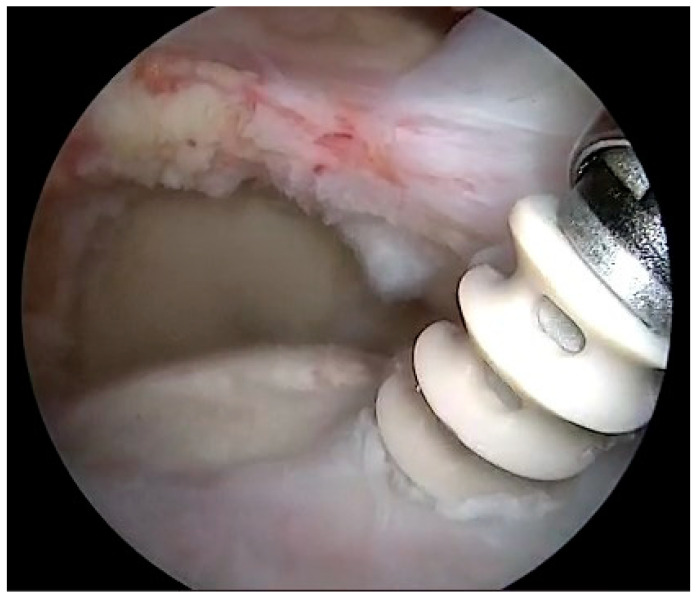
Right shoulder: scope in the lateral portal. Anchor placement. PEEK knotted suture anchors double-loaded with #2 high-strength sutures were placed at the medial edge of the tendon footprint.

**Figure 3 jcm-12-05694-f003:**
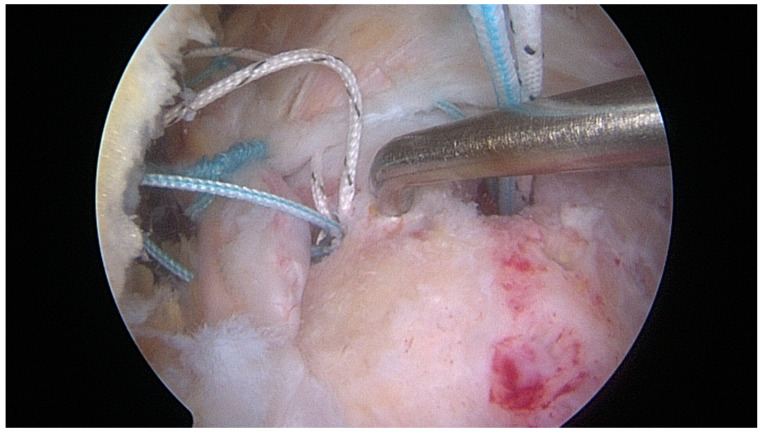
Right shoulder: scope in the lateral portal. After anchor placement and suture passage and before knot tying, multiple microfractures were performed using an angled arthroscopic awl for small joints onto the footprint area, between and just lateral to the suture anchors.

**Figure 4 jcm-12-05694-f004:**
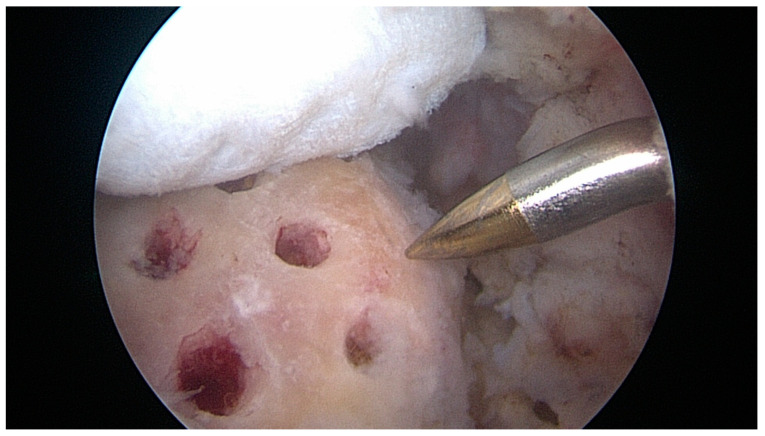
Right shoulder: scope in the posterior portal. After knot tying, all the remaining exposed area of the greater tuberosity was vented with multiple microfractures. The ECM patch graft was then applied on top to cover the entire area.

**Figure 5 jcm-12-05694-f005:**
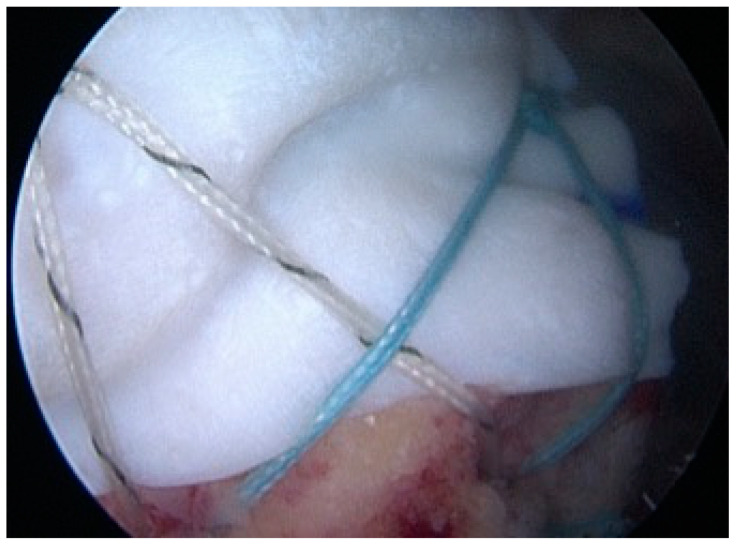
Right shoulder: scope in the lateral portal. Final view. The ECM patch graft was fixed over the RCR in a suture bridge configuration.

**Table 1 jcm-12-05694-t001:** Demographic data of the study population.

Variables		Overall
		N = 40
Gender	Male, N (%)	18 (45%)
	Female, N (%)	22 (55%)
Dominance	No, N (%)	10 (25%)
	Yes, N (%)	30 (75%)
Type of work	Manual, N (%)	18 (45%)
	Sedentary, N (%)	22 (55%)
Tear size	Large, N (%)	29 (72.5%)
	Massive, N (%)	11 (27.5%)

**Table 2 jcm-12-05694-t002:** Differences between groups for baseline characteristics.

Variables		Controls	Augment	*p*
		N = 20	N = 20	
Age (years)	Mean ± SD	63.8 ± 7.3	63.3 ± 7.5	0.832
Gender	Male, N (%)	10 (50%)	8 (40%)	0.376
	Female, N (%)	10 (50%)	12 (60%)	
Dominance	No, N (%)	4 (20%)	6 (30%)	0.358
	Yes, N (%)	16 (80%)	14 (70%)	
Type of work	Manual, N (%)	7 (35%)	11 (55%)	0.170
	Sedentary, N (%)	13 (65%)	9 (45%)	
Tear size	Large, N (%)	16 (80%)	13 (65%)	0.240
	Massive, N (%)	4 (20%)	7 (35%)	
Follow-up (months)	Mean ± SD	36.2 ± 9.1	36.1 ± 8.6	0.986
Constant score	Mean ± SD	47.9 ± 12.7	48.7 ± 18.6	0.873
Quick-DASH	Mean ± SD	59.8 ± 18.3	57.4 ± 9.9	0.610

**Table 3 jcm-12-05694-t003:** Differences between groups for clinical and structural outcomes.

Variables		Controls	Augment	*p*
		N = 20	N = 20	
Constant score	Mean ± SD	80.7 ± 16.6	91.5 ± 11.5	0.022 *
Quick-DASH	Mean ± SD	28.6 ± 21.6	20.1 ± 17.4	0.178
Tendon healing	Yes, N (%)	6 (30%)	16 (80%)	0.002 *
	No, N (%)	14 (70%)	4 (20%)	

* Statistically significant difference.

## Data Availability

The data presented in this study are available on request from the corresponding author.

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
