# Peer review of "Combined Biologic Augmentation Strategies with Collagen Patch Graft, Microfractures, and Platelet Concentrate Injections Improve Functional and Structural Outcomes of Arthroscopic Revision Rotator Cuff Repair"

_jcm, 2023, doi:10.3390/jcm12175694_

Round 1

Reviewer 1 Report

Introduction

1. I am missing a perspective regarding a research background at the end of this section. I would recommend adding a brief perspective before the aim. Then, it should be a clear message for the reader.

2. According to the previous comment, add also some reference and statments due to your hypothesis.

Materials and methods

1. How many patients were evaluated in this study? Add demographic and anthropometric information, with inclusion and exclusion criteria.

Conclusion

1. Please add some conclusions to this section. The statement you used is an observation and not a conclusion.

Author Response

Introduction

  1. I am missing a perspective regarding a research background at the end of this section. I would recommend adding a brief perspective before the aim. Then, it should be a clear message for the reader.

Authors: done, please see lines 55-60

  1. According to the previous comment, add also some reference and statements due to your hypothesis.

 Authors: done, please see lines 57-60

Materials and methods

  1. How many patients were evaluated in this study? Add demographic and anthropometric information, with inclusion and exclusion criteria.

Authors: done, please see lines 82-101

Conclusion

  1. Please add some conclusions to this section. The statement you used is an observation and not a conclusion.

Authors: done, please see lines 334-335

Reviewer 2 Report

Thanks for  letting me review the current manuscript  which conclude that:” Biologic augmentation of revision RCR by combination of microfracture, collagen 297 patch graft and subacromial injections of platelet concentrate is an effective strategy in 298 improving tendon healing rate.”.

Interesting manuscript, I do however have some concerns, which could have improve the manuscript.

1.       As MRI was conducted, it should have been with contrast to investigate the possibility of leakage through the cuff.

2.       There is little information on the retear rate in the control group, as written in the manuscript this is found to reduce the numbers of re-tears. I would  like to know how the result in the control group match those published previously to ensure that the comparison between the groups are based on correct para meters.

no comments

Author Response

  1. As MRI was conducted, it should have been with contrast to investigate the possibility of leakage through the cuff.

Authors: a new paragraph was added in the discussion (please see lines 316-324)

  1. There is little information on the retear rate in the control group, as written in the manuscript this is found to reduce the numbers of re-tears. I would  like to know how the result in the control group match those published previously to ensure that the comparison between the groups are based on correct para meters.

Authors: please see line 302-304

Round 2

Reviewer 1 Report

I accept paper in its current form.